# Drugs for Intermittent Preventive Treatment of Malaria in Pregnancy: Current Knowledge and Way Forward

**DOI:** 10.3390/tropicalmed7080152

**Published:** 2022-07-28

**Authors:** Antia Figueroa-Romero, Clara Pons-Duran, Raquel Gonzalez

**Affiliations:** 1Barcelona Institute for Global Health (ISGlobal), Hospital Clinic-Universitat de Barcelona, Carrer Rosselló 132, 08036 Barcelona, Spain; antia.figueroa@isglobal.org (A.F.-R.); cponsduran@hsph.harvard.edu (C.P.-D.); 2Centro de Investigação em Saúde de Manhiça (CISM), Manhiça, Maputo 1929, Mozambique; 3Consorcio de Investigación Biomédica en Red de Epidemiología y Salud Pública (CIBERESP), 28029 Madrid, Spain

**Keywords:** malaria, pregnancy, prevention, drugs, HIV

## Abstract

Malaria infection during pregnancy is an important driver of maternal and neonatal health in endemic countries. Intermittent preventive treatment in pregnancy (IPTp) with sulfadoxine-pyrimethamine (SP) is recommended for malaria prevention at each scheduled antenatal care visit, starting at the second trimester, in areas of high and moderate transmission. However, the increased resistance to SP in some endemic areas challenges its effectiveness. Furthermore, SP is contraindicated in the first trimester of pregnancy and in HIV-infected women on co-trimoxazole prophylaxis due to potential drug–drug interactions. Thus, in recent last decades, several studies evaluated alternative drugs that could be used for IPTp. A comprehensive literature review was conducted to summarize the evidence on the efficacy and safety of antimalarial drugs being evaluated for IPTp. Chloroquine, amodiaquine, mefloquine and azithromycin as IPTp have proven to be worse tolerated than SP. Mefloquine was found to increase the risk of mother-to-child transmission of HIV. Dihydroartemisin-piperaquine currently constitutes the most promising IPTp drug alternative; it reduced the prevalence of malaria infection, and placental and clinical malaria in studies among HIV-uninfected women, and it is currently being tested in HIV-infected women. Research on effective antimalarial drugs that can be safely administered for prevention to pregnant women should be prioritized. Malaria prevention in the first trimester of gestation and tailored interventions for HIV-infected women remain key research gaps to be addressed.

## 1. Introduction

### 1.1. Global Distribution of Malaria

Despite progress made in recent decades, malaria constitutes one of the most important poverty-related diseases in the world, accounting for nearly 627,000 deaths worldwide in 2020 [1]. It is estimated that nearly half of the global population is at risk of contracting malaria, sub-Saharan Africa (SSA) being the region that bears the brunt of the disease. Of note, the World Health Organization (WHO) African Region accounted for 95% of all cases in 2020 [1].

Malaria is caused by Plasmodium parasites, which are transmitted by the bite of an infected female Anopheles mosquito [2]. *P. falciparum* is the predominant and most virulent, and concentrates in the WHO African, South-East Asia, Eastern Mediterranean and Western Pacific Regions, while *P. vivax* is predominant in the WHO Region of the Americas [1,3].

### 1.2. Burden, Effects and Clinical Manifestations of Malaria in Pregnancy

In 2020 in 33 African countries with moderate and high malaria transmission, there were an estimated 33.8 million pregnancies, of which 11.6 million (34%) were exposed to malaria infection. Furthermore, it is estimated that 819,000 children were born with low birthweight in 2020 as a consequence of malaria in pregnancy [1]. It is estimated that *P. falciparum* causes 50,000 maternal deaths and 200,000 stillbirths each year [1].

In high-transmission areas, pregnant women have an increased susceptibility to malaria infection due to immunological and hormonal changes, placental sequestration of the parasite and an increased mosquito attraction [4]. The severity of malaria effects in pregnancy depends on the level of pre-acquired antimalarial immunity of women and other factors such as the number of previous pregnancies exposed to the infection, being primigravidae at increased risk. In turn, acquired immunity depends on the level of malaria transmission of the residence area [5,6].

Despite asymptomatic malaria infections being common, pregnant women suffer more symptomatic infections than their non-pregnant counterparts [7]. The effects of malaria have been extensively described on maternal, fetal and infant’s health. On pregnant woman, malaria can cause maternal anemia, severe disease, hypoglycaemia, placental infection and death [5,8,9]. The effects on the fetus and newborn range from congenital malaria, fetal anemia and increased risk of and adverse pregnancy outcomes such as miscarriage, stillbirth and preterm birth. In addition, placental infection can lead to intrauterine growth retardation and low birthweight, which in turn can lead to child growth retardation and poor cognitive and behavioral development [5,8,9].

HIV infection affects pregnancy-specific immunity, resulting in an increased susceptibility of HIV-infected women to anaemia, severe malaria, placental malaria and maternal death due to malaria infection [10,11,12,13]. HIV and malaria co-infection is also associated with an increased risk of low birthweight and preterm delivery compared with women with either infection alone [10,13]. In addition, the specific malaria immunity developed with subsequent malaria-exposed pregnancies is lost in HIV-infected women, shifting malaria burden from primi- and secundigravidae towards all gravidae women [14].

### 1.3. Current Recommendations for Malaria Prevention in Pregnancy

Currently, the WHO recommends the administration of intermittent preventive treatment in pregnancy with sulfadoxine-pirymethamine (IPTp-SP) at each scheduled antenatal care (ANC) visit, starting in the second trimester of gestation, with doses administered at least one month apart for all women living in moderate to high transmission areas [15]. The goal is to ensure the administration of at least three doses of SP during pregnancy [15]. Sulfadoxine-pirymethamine (SP) is an antifolate drug with a good safety profile in pregnancy, and it is currently the only recommended option for IPTp among HIV-uninfected women.

Each course of IPTp-SP consists on the oral administration of three tablets, each containing 500 mg of sulfadoxine and 25 mg of pyrimethamine, irrespective of the presence of parasites, and under directly observed therapy (DOT) [15]. IPTp-SP has been proven to reduce maternal and placental infection and rates of miscarriage [16,17]. In addition, it reduces low birth weight (LBW), maternal and neonatal mortality, and protects against maternal and foetal anaemia [17,18,19].

Despite the efficacy and safety of IPTp-SP, several factors challenge the effectiveness of the strategy. Firstly, following the widespread adoption of IPTp-SP in SSA, there has been increased resistances to this drug by *P. falciparum* due to mutations in dihydrofolate reductase (dhfr) and dihydropteroate synthase (dhps) genes [20,21,22]. In addition, SP is not recommended for women who are in their first trimester of pregnancy due to possible teratogenic effects and limited evidence in humans [15,23], leaving these women unprotected against malaria infection and its deleterious effects [22]. Finally, SP uptake is contraindicated in HIV-infected women on co-trimoxazole prophylaxis (CTXp) due to potential drug–drug interactions [22].

### 1.4. Review Justification and Search Limits

Because of the aforementioned challenges of IPTp-SP that may limit its effectiveness, several antimalarial drugs have been evaluated as potential candidates to replace SP for IPTp in recent decades. The main attributes of an antimalarial drug to be considered for IPTp are: (1) having an acceptable reproductive toxicity profile in pregnancy; (2) long half-life to be administered ideally at the time of the monthly ANC clinic visits; (3) single-day administration (to ensure DOT); (4) affordable; and (5) acceptable to pregnant women [24]. Of note, pregnant women are often excluded from drug trials for ethical and legal issues, as well as for safety concerns [25].

This review examines the available information on the safety and efficacy of the alternative antimalarial drugs that have been evaluated for the prevention of malaria in pregnancy. A comprehensive literature search of medical databases (Pubmed, WHO) and non-medical search engines was conducted looking for published studies reporting on the efficacy and safety of antimalarials drugs that have been evaluated for IPTp using the following keywords: pregnancy, malaria, antimalarial, control, prevention, drugs, efficacy and safety. A summary of the characteristics and main finding of the retrieved studies is provided in Table 1.

## 2. Drugs for Prevention of Malaria in Pregnancy

### 2.1. Amodiaquine

Amodiaquine (AQ) is a 4-aminoquinoline drug with antipyretic and anti-inflammatory properties [41]. It has been effective when used for perennial malaria chemoprevention and for the treatment of uncomplicated malaria in pregnant women [42,43].

Efficacy: A three-arm randomized clinical trial (RCT) performed in Ghana in 2007 that compared IPTp-SP (single dose), IPTp-AQ (full treatment course), and IPTp-SP + AQ given over three days concluded that prevalence of anemia, LBW and peripheral and placental parasitaemia did not differ significantly between arms [26].

Safety/tolerability: The aforementioned study found that Ghanaian women who received AQ or SP+AQ were more likely to report adverse events than those who received SP, including body pains, weakness, dizziness, vomiting and nausea [26]. In addition, AQ can cause toxic hepatitis and agranulocytosis in the general population, although evidence from a Phase IV implementation study to assess the real-life safety of SP+AQ among children is reassuring [44,45]. Given the increased frequency of adverse events, IPTp-AQ both in mono and combination therapy is unlikely to be useful as an alternative to IPTp-SP.

### 2.2. Chloroquine

Chloroquine (CQ) is a 4-aminoquinoline drug that was considered the gold standard for the prevention of malaria as weekly prophylaxis prior to the increase in CQ resistance in the late 1990s [46]. The drug is also used in pregnancy for the treatment of inflammatory rheumatic diseases [47].

Efficacy: Prior to the development of high rates of resistance, CQ had been shown to reduce placental infection, improved maternal hemoglobin and increased birth weight [48,49,50]. Since 2006, results from studies performed in Malawi, Tanzania, Zambia and Senegal indicate that sensitivity to CQ is increasing again, and thus the drug was evaluated for IPTp in an RCT in Malawi in 2018 [37,51,52,53,54]. This trial found that weekly prophylaxis or four doses of IPTp-CQ were not more effective than IPTp-SP in reducing placental malaria [37]. However, when adjusting for potential predictors of placental malaria, the reduction in placental malaria in weekly CQ recipients was statistically significant [37].

Safety/tolerability: CQ has well established dosing and safety profiles in pregnancy, including during the first trimester of gestation [55]. However, CQ resistance, together with poor adherence in the case of weekly administration, has limited its use for malaria chemoprevention [17,56]. In addition, CQ use was associated to an increased risk of non-severe adverse events such as dizziness and vomiting compared to IPTp-SP in the trial conducted in Malawi [37].

### 2.3. Mefloquine

Mefloquine (MQ) is a quinolinemethanol with antiparasitic properties. Since its development, it has been indicated for both prophylaxis and treatment of malaria [44]. MQ has been considered an optimal candidate for IPTp because it has a long half-life in serum (median between 12 and 17 days at prophylactic doses), simplicity of administration (single dose), low associated rates of *Plasmodium* resistance in SSA, as well as a well characterized pharmacokinetic profile in pregnant women and an acceptable reproductive toxicity profile in animal studies [24,57]. MQ is recommended by the WHO and the US Centers for Disease Control (CDC) for pregnant women of all gestational ages travelling to malaria-endemic regions [58].

Efficacy: A RCT performed in Benin between 2005 and 2008 that compared two doses of IPTp-SP with two doses of IPTp-MQ [27], and another trial performed in Benin, Gabon, Tanzania and Mozambique between 2009 and 2013 [28], which compared two doses of IPTp-SP with a single dose of MQ (15 mg/kg) or split doses (7.5 mg/kg given in two consecutive days), found that IPTp-MQ was not more effective than IPTp-SP in reducing the incidence of LBW and other maternal and infant outcomes [27,28,59]. However, IPTp-MQ was more effective than IPTp-SP in decreasing maternal anaemia and reducing maternal peripheral parasitaemia at delivery [27,28,59]. No differences in the prevalence of adverse pregnancy outcomes (miscarriage, stillbirth and congenital malformations) were found between IPTp-SP and IPTp-MQ [27,28,59]. Similarly, no differences were found between arms on infant’s outcomes (mortality, underweight, incidence of clinical malaria, and nutritional outcomes) after one year of follow up [28,60].

Safety/tolerability: These studies found poorer tolerability of MQ compared to SP. Furthermore, a Cochrane review including both RCTs, reported a four-fold increase in vomiting and dizziness among MQ recipients compared to SP, as well as tiredness and nausea [27,28,59]. Of note, splitting MQ dose administration over two days did not improve its tolerability [28]. In addition, although rare, neuropsychiatric side effects of MQ have been reported after use of MQ as chemoprophylaxis in an adult population [61]. Considering the reported drug-related adverse effects, MQ has not been recommended for IPTp by the WHO [62].

### 2.4. Dihydroartemisinin-Piperaquine

Artemisinin-based combination therapies (ACT) are currently recommended for the treatment of uncomplicated malaria in pregnancy in second and third trimesters of gestation [63]. Dihydroartemisinin-piperaquine (DP) is an ACT that is currently been evaluated for IPTp. The long half-life of piperaquine in serum (about 23 days in adults), the fact that it is not usually used as first-line malaria treatment and its good tolerability make DP a good candidate for IPTp [64,65,66,67].

Efficacy: To date, four randomized clinical trials conducted in Kenya, Uganda and Tanzania have evaluated DP as IPTp [32,35,36,68]. These studies found that prevalence of malaria infection, placental malaria and clinical malaria were lower in the IPTp-DP arm than in the IPTp-SP group [32,35,36,68]. The studies performed in Kenya and Uganda found that the risk of stillbirths and infant mortality within 6-8 weeks after birth were decreased among women in the DP group, but this finding was not confirmed by the study performed in Tanzania, which found that the prevalence of adverse birth outcomes did not differ significantly between groups [36]. Moreover, the trials performed in Kenya and Uganda were underpowered to detect differences in the risk of adverse pregnancy outcomes [32]. Additionally, no differences were found in the incidence of malaria in infants born to IPTp-DP recipients compared to those born to IPTp-SP-recipient women [69]. However, when stratifying by infant sex, the incidence of malaria was significantly lower among male infants born to mothers who received IPTp-DP compared to those born to mothers who received IPTp-SP [70].

Safety/tolerability: DP has not shown differences in the frequency of drug-related adverse events compared with SP [32,35,36,68], with usually mild and transient adverse effects, including dizziness, nausea, vomiting, and headache [36,71,72,73]. Non-severe dysphagia has been reported more frequently by women on monthly DP than those on three-dose DP [32]. In addition, piperaquine has been associated with a dose-dependent QT interval prolongation [72,73,74], also among pregnant women. Consequently, its use is contraindicated for patients at risk of QTc interval prolongation or cardiac arrhythmias, or those who are taking other QT prolonging drugs [73].

### 2.5. Azithromycin

Azithromycin (AZ) is a semisynthetic azalide with a broad spectrum of antibacterial activity and a relatively long half-life in serum (between 8 and 24 h) [75]. AZ has been used for the treatment and prevention of yaws and sexually transmitted infections such as *Neisseria gonorrheae* [76,77,78].

Efficacy: Its antimalarial activity is relatively weak. A trial performed in Malawi that compared IPTp-AZ (two tablets per day, at 16–24 and 28–32 week of gestation) with placebo did not find differences between groups in the frequency of preterm birth, mean gestational age at delivery, mean birthweight, perinatal deaths, maternal malarial parasitaemia and anaemia [79]. These results are against recommending AZ for IPTp. In addition, concerns on increasing AZ resistance must be taken into account when considering AZ for IPTp [80]. For instance, high levels of AZ resistance have been reported in Ethiopia following widespread AZ mass administration for non-chlamydia conjunctival bacteria since 2003 [81].

Safety/tolerability: AZ has been demonstrated to be safe over the course of pregnancy [82]. Adverse effects of AZ are commonly mild and mainly gastro-intestinal, including diarrhea, nausea and vomiting. Dizziness and headache have also been associated with AZ uptake [83].

AZ has been tested as chemoprophylaxis in combination with other antimalarial drugs:

#### 2.5.1. Azithromycin + Chloroquine

Efficacy: A trial performed in five African countries that compared monthly 3-day IPTp with AZ 1 g and CQ 620 mg to IPTp-SP was terminated early due to futility [40]. This trial found that despite AZ + CQ being safe and reducing the incidence of symptomatic malaria and peripheral parasitemia at weeks 36 to 38 of gestation, it was not more effective than SP in decreasing the risk of adverse pregnancy outcomes, such as LBW, stillbirth and abortion [40].

Safety/tolerability: Adverse events were more commonly reported in the AZ + CQ. Those were mild and included vomiting, dizziness, headache, and asthenia [40].

#### 2.5.2. Azithromycin + Sulfadoxine/Piperaquine

Efficacy: A trial performed in Malawi that compared two doses of IPTp-SP with monthly IPTp-SP plus two doses of AZ found that infants born to participants who took monthly IPTp-SP + AZ had a 40% decreased incidence of LBW, increased mean duration of their pregnancy (0.4 weeks) and decreased prevalence of malaria parasitemia [39]. In addition, compared to participants who took two doses of IPTp-SP, women in the monthly IPTp-SP plus two doses of AZ had a 77% decrease in prevalence of PCR-diagnosed *P. falciparum* malaria parasitaemia at delivery [84]. However, it is not clear if the added efficacy of the IPTp-SP + AZ regimen was due to the effect of AZ against reproductive tract infections (RTI) rather than to an antimalarial effect [84]. Infants born to women in the AZ-SP group weighted on average 140 g more at birth and were 0.6 cm longer at four weeks of age compared to infants born to women in the two doses of IPTp-SP, probably due to the activity of AZ against RTI [85].

Outside Africa, a trial carried out in Papua New Guinea between 2009 and 2013 compared IPTp-SP with AZ (one dose daily for 2 days) both given three times, with one course of SP and CQ (three or four tablets (150 mg)), daily for 3 days [86]. IPTp-SP with AZ significantly reduced the risk of LBW and preterm delivery, and increased mean birthweight compared to a single treatment course of SP and CQ. In addition, mean birthweight was 41.9 g higher in the intervention arm [86]. Women receiving the intervention were at lower risk of peripheral and placental blood parasitaemia as well as active placental infection. However, there was no significant difference in the proportion of women with anaemia at delivery. In terms of adverse pregnancy outcomes, no significant differences in the proportion of maternal deaths, miscarriages, stillbirths, congenital abnormalities, and neonatal deaths between both arms were observed [86]. A subsequent sub-study concluded that IPTp-SP with AZ may protect against adverse pregnancy outcomes by reducing inflammation and preventing its deleterious consequences [87].

Safety/tolerability: Common reported side effects of AZ + SP include vomiting, dizziness, nausea, itching, weakness, and abdominal pain. Women receiving IPTp-SP + AZ experienced dizziness and abdominal pain less frequently than women who had IPTp-SP with CQ [86].

#### 2.5.3. Azithromycin + Piperaquine

Efficacy: An open-label trial performed among 122 pregnant women in Papua New Guinea compared three daily doses of AZ-piperaquine with SP and concluded that there was no difference in blood smear positivity rates between AZ-piperaquine and SP by the time of delivery and up to day 42 [88].

Safety/tolerability: AZ-piperaquine was proved to be safe and associated with mild but frequent adverse effects, including nausea, dizziness, vomiting, and abdominal pain [88,89].

### 2.6. IPTp for HIV-Infected Pregnant Women

Globally, in 2020, there were 1.3 million pregnant women with HIV, of which an estimated 85% received antiretroviral drugs [90]. HIV-infected pregnant women are known to be at increased risk of malaria and its adverse maternal consequences [12]. Universal cotrimoxazole prophylaxis (CTXp) is currently recommended regardless of CD4 + T count to prevent opportunistic infections among HIV-infected pregnant women living in areas with limited health resources and high HIV prevalence [91]. CTXp is a fixed-dose drug combination of trimethoprim and sulfamethoxazole. This prophylactic treatment has proved to reduce the risk of malaria in adults and children in SSA [92,93], as well as in pregnant women [94].

Importantly, IPTp with SP is contraindicated in HIV-infected pregnant women who take CTXp due to possible sulfonamide-induced adverse drug reactions such as Stevens-Johnson syndrome, erythema multiforme and leucopenia [22,91,95,96]. Consequently, the women most vulnerable to malaria, those who are HIV infected and pregnant, cannot receive IPTp [97].

A trial performed in Malawi among 264 HIV-infected women that compared three-dose IPTp-SP with daily CTXp failed to find daily CTXp to be non-inferior for preventing maternal infection given that the number of cases of clinical malaria (trial’s primary outcome) was not different among groups [98]. CTXp was found to be safe and to have a similar efficacy as SP for preventing the prevalence of parasitemia, placental malaria and adverse neonatal outcomes such as prematurity, stillbirths and LBW [98]. Additionally, another study performed in Zambia among HIV-infected and HIV-uninfected women showed that exposure to daily CTXp is safe for mothers’ and newborns’ health [99].

#### 2.6.1. Mefloquine

Efficacy: The efficacy and safety of MQ for IPTp have been evaluated in RCTs comparing it with daily CTXp or IPTp-SP for the prevention of malaria in HIV-infected pregnant women [29,30,31]. A clinical trial performed in Benin that compared IPTp-MQ versus daily CTXp could not provide conclusive results due to a too small sample size [30]. Another arm in the same trial compared the uptake of daily CTXp versus IPTp-MQ + CTXp, and found that placental parasitemia was decreased among women in the CTXp + IPTp-MQ group [30]. A multicenter placebo-controlled trial also evaluated IPTp-MQ among 1071 HIV-infected women on CTXp in Kenya, Mozambique and Tanzania [29]. This study found that prevalence of peripheral maternal malaria infection at delivery and placental infection were significantly lower in women receiving CTXp + IPTp-MQ compared to those who received CTXp alone [29]. In addition, there were no differences in the frequency of adverse pregnancy outcomes such as stillbirths and neonatal deaths between arms [29]. A more recent open-label trial among 131 HIV-infected pregnant women from Nigeria compared IPTp-MQ versus IPTp-SP and did not find differences in malaria outcomes across study arms.

Safety/tolerability: All studies have reported a poor tolerability of MQ. Dizziness, vomiting and nausea were increased in women receiving IPTp-MQ in the aforementioned placebo-controlled trial. Importantly, the same trial reported a two-fold increased risk of mother-to-child transmission (MTCT) of HIV among infants born to MQ recipients [29]. It has been hypothesized that this finding is due to a reduction in the concentration of nevirapine linked to the uptake of MQ [100].

#### 2.6.2. Dihydroartemisinin-Piperaquine

Efficacy: Monthly DP has also been tested for preventing malaria in pregnancy among HIV-infected pregnant women [34]. When comparing monthly IPTp-DP + CTXp versus CTXp alone, there was no difference in the risk of placental malarial infection or birth outcomes such as stillbirth and LBW between both arms, concluding that there is no additional benefit on adding IPTp-DP to the daily CTXp regimen [34]. However, authors acknowledged the limitations of their results, such as the small sample size and the low prevalence of malaria in the study area at the time of the trial [34].

Safety/tolerability: DP was well tolerated. There were no significant differences in the incidence of adverse events of any severity [34].

#### 2.6.3. Azithromycin

Efficacy: A study performed in Nigeria that compared the use of daily AZ versus monthly IPTp-SP among HIV-infected pregnant women found that both drugs are comparable in terms of efficacy since the prevalence of clinical malaria during pregnancy did not differ significantly among groups [38].

Safety/tolerability: In that trial, side effects were mild and uncommon, with the exception of nausea, which was significantly higher in the daily AZ arm [38].

### 2.7. Chemoprevention during the First Trimester of Pregnancy

It is estimated that the proportion of pregnant women who attend the ANC clinic before the 12th week of gestation in SSA countries is 38.0%, ranging from 14.5% in Mozambique to 68.6% in Liberia [101]. Timely initiation of ANC constitutes a basic component of ANC services that helps to prevent health conditions such as malaria. Of note, there is evidence that malaria infection during the first trimester of pregnancy is associated with an increased risk of LBW and maternal anemia [102,103,104]. However, SP is contraindicated during the first trimester of gestation given concerns on its teratogenic effects, and thus prevention of malaria during the first trimester of pregnancy lays on the use of long-lasting insecticide-treated nets (LLITNs) [15]. The prevalence of malaria in first trimester of gestation was estimated to be of 22% in a cohort of Beninese women [102]. Antimalarial drugs that can be administered safely in first trimester of pregnancy are MQ and CQ. MQ is recommended for pregnant women travelling to malaria-endemic areas as chemoprophylaxis, including those in the first trimester of gestation [58]. With regard to CQ, given the return of CQ-susceptible malaria in some areas of southern Africa, it could be reconsidered for malaria chemoprevention in the first trimester of pregnancy [51,52,53,54].

## 3. Discussion and Way Forward

Malaria infection, and specifically *P. falciparum*, remains a threat to maternal and neonatal health in SSA. IPTp-SP has significantly reduced the burden and impact of malaria in pregnancy in endemic settings, as it is associated with beneficial effects on maternal and fetal health outcomes, mainly reductions in LBW [33]. In addition, it is still the most cost-effective intervention for the prevention of malaria in pregnancy in SSA [105]. However, the protective effect of IPTp-SP is threatened by the presence of resistance mutations that compromise the clearance of parasites in pregnant women [106].

Several drugs have been evaluated as potential alternatives for IPTp, although most of them were discarded due to limited increased efficacy over SP or safety concerns. For instance, MQ showed poor tolerability with frequent dizziness and vomiting. Despite a proven reduction in resistances to CQ during recent years, IPTp-CQ was not more effective than IPTp-SP against malaria infection during pregnancy, while it was associated to an increased risk of adverse events such as dizziness and vomiting [37]. However, when adjusting for potential predictors of placental malaria, the reduction in placental malaria in weekly CQ recipients was statistically significant, suggesting that CQ remains a valuable alternative to IPTp-SP worthy of future research [37]. AZ’s efficacy for IPTp is weak [38]. The combination of AZ with CQ yielded poor results, and the combination with piperaquine did not show any advantages in comparison to IPTp-SP [40,89]. IPTp-SP + AZ was well tolerated and efficacious among HIV-uninfected women [39,84]. However, it is not clear if the protective efficacy was linked to its antimalarial effect but rather to the effect of AZ against RTI [39,84]. In addition, this combination would potentially increase drug pressure and accelerate the emergence of SP-resistant mutations in *P. falciparum.*

DP constitutes a promising candidate for IPTp. Among HIV-uninfected women, it reduced prevalence rates of malaria infection, placental malaria and clinical malaria compared to IPTp-SP. However, DP was not superior to SP in reducing the risk of LBW; the broad antibacterial effect of SP might explain these differences. Similarly, the fact that CTXp has been found to have a similar efficacy as SP for preventing adverse neonatal outcomes such as LBW also supports the hypothesis of the broader anti-bacterial effect of SP.

Two clinical trials in Mozambique, Gabon, Kenya and Malawi are currently being conducted to evaluate the safety, tolerability and efficacy of DP as IPTp for malaria prevention in HIV-infected pregnant women receiving daily CTXp and ARV drugs [107,108].

Despite its promising characteristics, DP is not without challenges. First, the use of ACTs for preventive treatment should be considered with caution since it can contribute to the spread of artemisinin-resistant *P. falciparum* in Africa [109]. Given that ACTs are currently the first line of malaria treatment, the spread of resistant parasites poses a great concern and highlights the need to evaluate whether it is necessary to restrict the use of ACTs for the treatment of malaria. Secondly, DP regimens, as other ACTs, is a three-day treatment course and thus directly observed therapy will not be feasible, potentially hindering a good adherence to the treatment. Finally, drug costs should be considered as a potential barrier to real-world implementation. For instance, the Global Fund price reference per treatment for SP is USD 0.28, while for DP it is around USD 1.50, depending on the formulation and pack sizes [110]. In areas of high SP resistance and high malaria transmission, the use of DP for IPTp is cost-effective for HIV-uninfected pregnant women with high uptake of long-lasting insecticidal nets, but currently no information on cost-effectiveness is available for HIV-infected women [111].

Bearing in mind the challenge of increasing resistances, other ACTs such as pyronaridine-artesunate (PA), which is currently being tested for intermittent screening and treatment [112], could be considered for IPTp if PA demonstrates to be effective and safe for pregnant women.

Metronidazole, a nitroimidazole with anti-microbial and anti-protozoan activity, is being considered for IPTp in combination with SP or in combination with DP in a currently ongoing RCT in Zambia [113]. Investigators hypothesize that the combination of an antimalarial with a drug against sexually transmitted infections may produce better birth outcomes than an antimalarial in monotherapy [113,114]. Despite its potential, metronidazole resistances will need also to be considered should this combination be effective for the prevention of malaria in pregnancy [115].

Besides IPTp, other control tools must be considered. The RTS,S vaccines have been developed and approved for the prevention of malaria in children living in regions with moderate to high malaria transmission [116]. Yet, pregnant women still lack a malaria vaccine. Promising vaccine candidates for the prevention of malaria in pregnancy include the PRIMVAC and PAMVAC vaccines, which have shown an acceptable safety profile and have been found to induce functional antibodies in phase I clinical trials performed among healthy non-pregnant adults. [117,118]. CIS43LS human monoclonal antibodies are also being evaluated among malaria-naïve adults and could eventually be also safe and efficacious in pregnancy [119].

African HIV-infected pregnant women are the most vulnerable population to malaria infection [120], and, paradoxically, they are also the least protected due to the difficulties in finding an alternative to SP given potential interactions between antiretroviral and antimalarial drugs [121]. Those interactions could challenge the treatment and prevention of both infections, although most data come from in vitro studies and their clinical relevance is uncertain [97,121]. In the absence of alternatives to SP for IPTp, daily CTXp is the recommended malaria prevention therapy for HIV-infected pregnant women. Other prophylactic options for HIV-infected women that could be safely administered concomitantly with the current standard of care, CTXp and with ARV drugs, need to be evaluated.

Regarding chemoprevention in the first trimester of pregnancy, SP is not recommended leaving a high proportion of pregnant women in endemic areas pharmacologically unprotected and relying on the use of LLITNs. Among the antimalarial drugs considered safe during the first trimester of gestation, CQ might still provide protection in some areas of SSA and could be evaluated for chemoprevention [51,52,53,54].

## 4. Conclusions

Several antimalarial drugs have been evaluated for IPTp in recent decades. Most of them showed worse tolerability than SP, DP being one of the current most promising candidates. Although SP is still effective at preventing the deleterious effects of malaria in pregnancy, the spread of resistance and the limitations of its use in the first trimester of pregnancy and among HIV-infected women on CTXp warrant the evaluation of alternative drugs. ACTs such as DP are being considered as potential IPTp drugs, but recent reports of artemisinin resistance in Africa highlight the need to continue studying alternative compounds. Of note, pregnant women should be included in these evaluations and clinical trials to ensure high-quality evidence on safety and efficacy on this particularly vulnerable population [122]. Besides, the cost-effectiveness of all promising candidates needs to be evaluated before being recommended in endemic areas. Complementary approaches, especially during the first trimester of pregnancy, such as vaccines for pregnancy-associated malaria and monoclonal antibodies, are needed to reduce the burden caused by malaria [118,119]. Finally, HIV-infected pregnant women should also be prioritized in malaria research since currently they are the most vulnerable to malaria and, paradoxically, the least protected [97].

## Figures and Tables

**Table 1 tropicalmed-07-00152-t001:** Overview of the studies included in this review.

Drug	Study	Study Design	Study Year and Location	Malaria Indicators	Safety on Pregnancy Outcomes	Tolerability	Conclusion
AQ	[26] Clerk et al., 2008	Double-blind, three-arm RCT IPTp-AQ IPTp-AQSP IPTp-SP (N = 3643)	2004–2007 Ghana	- The prevalence of peripheral and placental malaria and anemia at delivery was similar groups.	- There was no difference between groups with regard to the incidences of LBW.	Women who received AQ or SPAQ were more likely to report adverse events than were those who received SP. Symptoms were usually mild, including bodily pains and weakness, dizziness, vomiting, and nausea.	- The effects of IPTp-AQ or SPAQ were comparable to the effects of IPTp-SP.
MQ	[27] Briand et al., 2009	Open-label equivalence RCT IPTp-MQ IPTp-SP (N = 1601)	2005–2008 Benin	- Placental and peripheral parasitemia at delivery were significantly less prevalent in the MQ group than in the SP group. - Women in the MQ group were less likely to have anemia than were women in the SP group, (difference only marginally significant)	- The incidences of spontaneous abortions, stillbirths, and congenital anomalies did not differ significantly between groups. - The prevalence of LBW among infants born to women receiving MQ and to women receiving SP was not statistically different.	- The proportion of women who reported an AE was significantly higher in the MQ group than in the SP group. The most common complaints were vomiting, dizziness, tiredness, and nausea.	MQ proved to be highly efficacious for use as IPTp. Its low tolerability might impair its effectiveness.
	[28] González et al., 2014	Open label, Three-arm, RCT IPTp-SP IPTp-MQ full dose IPTp-MQ split-dose (N = 4749)	2009–2013 Benin, Gabon, Tanzania, Mozambique	- IPTp-MQ was associated with lower rates of - Peripheral malaria parasitemia at delivery - Maternal anemia at delivery - Clinical malaria episodes - All-cause outpatient attendances during pregnancy - There were no differences between groups in the prevalence of - placental infection - neonatal parasitemia - neonatal anemia	- There were no significant differences between the MQ and SP groups in either the prevalence of LBW infants or in mean birth weight. - There was no difference in the prevalence of adverse pregnancy outcomes between groups, including miscarriages, stillbirths, and congenital malformations.	- The immediate tolerability of IPTp was poorer in the two MQ groups as compared to the SP group, with no difference between the MQ full and split-dose groups. The most frequently reported related AEs were dizziness and vomiting.	The results of this study do not support a change in the current recommended IPTp policy.
	[29] González et al., 2014	Double-blind two arm RCT: IPTp-Placebo + CTX IPTp-MQ + CTX (N = 1071 HIV-infected women)	2009–2013 Kenya, Tanzania and Mozambique	IPTp-MQ was associated with reduced rates of - maternal parasitemia - placental malaria - hospital admissions HIV viral load at delivery was higher in the MQ group compared to the control group.	- There were no differences in the prevalence of adverse pregnancy outcomes between groups. - The mother-to-child transmission of HIV was twofold higher in the IPTp-MQ group.	Drug tolerability was poorer in the MQ group compared to the control group (dizziness and vomiting after the first IPTp-MQ administration).	Its potential for IPTp is limited given poor drug tolerability and given that MQ was associated with an increased risk of mother-to-child transmission of HIV.
	[30] Denoueud-Ndam 2014 et al. (I)	Open label, non-inferiority RCT daily CTX daily CTX + IPTp-MQ (N = 140 HIV-infected women)	2009–2011 Benin	- CTX efficacy for the prevention of placental parasitemia was not more than 5% inferior to the association of CTX + MQ-IPTp. - No differences were either observed regarding peripheral parasitemia at delivery and maternal hemoglobin between groups.	No statistically significant differences were either observed regarding birth weight, or prematurity.	Vomiting, nausea, dizziness, and fatigue were more frequently reported with MQ.	Small sample size MQ-IPTp may be an effective alternative given concern about parasite resistance to CTX
	[30] Denoueud-Ndam 2014 et al. (II)	Open label, non-RCT daily CTX IPTp-MQ (N = 292 HIV-infected women)	2009–2011 Benin	Because of the small sample size obtained, noninferiority could not be conclusively assessed. No statistically significant differences were observed regarding peripheral parasitemia at delivery and maternal hemoglobin	No statistically significant differences were either observed regarding birth weight, or prematurity.	Vomiting, nausea, dizziness, and fatigue were more frequently reported with MQ.	MQ-IPTp may be an effective alternative given concern about parasite resistance to CTX
	[31] Akinyotu et al., 2018	Open label RCT IPTp-MQ IPTp-SP (N = 142 HIV-infected women)	2016 Nigeria	- Presence of malaria parasites in peripheral blood at delivery or enrolment.	No statistically significant differences were found in the incidence of preterm birth and LBW.	There was no significant difference in the occurrence of vomiting, gastric pain, headache and dizziness. Nausea was eight times more likely to occur in the MQ group.	Outcomes following use of IPTp-PQ were comparable to IPTp-SP treatment. The authors concluded that MQ is a feasible alternative therapy.
DP	[32] Kakuru et al., 2016	Three-arm, double-blind, RCT IPTp-SP 3-dose IPTp-DP or monthly IPTp-DP (N = 300)	2014 Uganda	- The prevalence of placental malaria was significantly higher in the SP group than in the three-dose DP group or the monthly DP group. - During pregnancy, the incidence of symptomatic malaria was significantly higher in the SP group than in the three-dose DP or the monthly DP.	- The prevalence of a composite adverse birth outcome was lower in the monthly DP group than in the SP group or the three-dose DP group.	- In each treatment group, the risk of vomiting after administration of any dose of the study agents was very low. - There were no significant differences among the groups in the risk of adverse events.	IPTp-DP during pregnancy resulted in a lower burden of malaria than did treatment with SP.
	[33] Desai et al., 2016	Three-arm, open-label RCT IPTp-DP IPTp-SP IST-DP (N = 1546)	2012–2014 Kenya	- Compared with women who received IPTp-SP, prevalence of malaria infection at the time of delivery was lower in the IPTp-DP group - Women in IPTp-DP group had - fewer malaria infections - lower incidence of clinical malaria - fewer all-cause sick-clinic visits during pregnancy than those in the IPTp-SP group	Women in the IPTp-DP group had fewer stillbirths, and infant mortality than those in the IPTp-SP group. Prevalence of LBW, small for gestational age, and preterm delivery did not differ significantly between groups.	- DP was well tolerated by most women. Adverse events were more frequent in the IPTp-DP group.	DP is a promising alternative drug to replace SP for IPTp.
	[34] Natureeba et al., 2017	Double-blinded, RCT daily CTX + monthly DP daily CTX (N = 200 HIV-positive women)	2014–2015 Uganda	No statistically significant difference in - risk of placental malarial infection - incidence of malaria and parasite prevalence among both arms.	No statistically significant difference in the incidence of adverse birth outcomes among both arms.	There were no significant differences in the incidence of adverse events of any severity.	Adding monthly DP to daily CTX did not reduce the risk of placental or maternal malaria or improve birth outcomes.
	[35] Kajubi et al., 2019	Double-blind, RCT IPTp-SP IPTp-DP (N = 782)	2016–2017 Uganda	IPTp-DP was associated with lower: - incidence of symptomatic malaria during pregnancy. - prevalence of parasitaemia at the time of each routine visit. - risk of maternal anaemia during pregnancy	- There was no significant difference in the risk of LBW, preterm birth, small for gestational age, or composite adverse birth outcome between the treatment groups.	- Both drug regimens were well tolerated, with no significant differences in adverse events between the groups, with the exception of asymptomatic corrected QT interval prolongation (significantly higher in the DP group).	Monthly IPTp-DP was safe but did not lead to significant improvements in birth outcomes compared with SP.
	[36] Mlugu et al., 2021	Open-label RCT IPTp-DP IPTp-SP (N = 956)	2017–2019 Tanzania	IPTp-DP was associated with lower: - prevalence of maternal malaria at delivery. - incidence of symptomatic-malaria and parasitemia during pregnancy	The prevalence of any adverse birth outcomes was not significantly different between groups. The prevalence of LBW was significantly lower in IPTp-DP.	There was no significant difference in the prevalence of adverse drug events between the treatment groups.	There was a significantly higher protective efficacy of IPTp-DP compared to monthly IPTp-SP.
CQ	[37] Divala et al., 2018	Three arm, open-label, RCT CQ-IPTp CQ chemoprophylaxis SP-IPTp (N = 900)	2012–2014 Malawi	There was no difference in the risk of - placental malaria detected by histopathology - malaria infection or clinical malaria illness.	There were no differences in adverse pregnancy outcomes between arms.	Both CQ treatment regimens were associated with higher rates of treatment-related adverse events than the SP-IPTp regimen.	This study did not have enough superiority evidence of chloroquine either as IPTp or as chemoprophylaxis versus SP-IPTp for prevention of malaria during pregnancy and associated maternal and infant adverse outcomes.
AZ	[38] Akinyotu et al., 2019	Open-label RCT IPTp-SP AZ (N = 180 HIV-infected women)	2015–2016 Nigeria	No statistically significant difference in the incidence of malaria parasitaemia at delivery and placental parasitization among arms.	No significant difference in preterm birth and LBW between both arms.	Nausea was significantly higher in the AZ group compared with the SP group. There were no statistically significant differences among groups in the presence of dizziness and headache.	The use of AZ for malaria prevention in HIV-positive pregnant women has a comparable outcome to SP. It is tolerable and has few maternal and foetal adverse effects
AZSP	[39] Luntamo et al., 2010	RCT Two-dose IPTp-SP Monthly IPTp-SP Monthly IPTp-AZSP (N = 1320)	2003–2006 Malawi	Compared with the controls, participants in the monthly SP and AZSP groups had a statistically significant lower prevalence of peripheral malaria parasitemia at 32 gestation weeks.	IPTp-SPAZ was associated with lower incidence of preterm delivery and LBWIPTp-SPAZ and monthly IPTp-SP were associated with higher mean duration of pregnancy.	Incidence of serious adverse events was low in all groups.	This intervention could be efficacious, but the impact would heavily depend on the local epidemiology and resistance of malaria.
AZCQ	[40] Kimani et al., 2016	Open-label RCT IPTp-SP IPTp-AZCQ (N = 2891)	2010–2013 Benin, Kenya, Tanzania, Uganda	Statistically significant reduction in - symptomatic malaria episodes - incidence of peripheral parasitemia at w. 36–38.	There was no significant difference in the incidence of LBW between treatment groups in the IPTp-AZCQ group.	AEs such as vomiting, dizziness, headache, and asthenia were reported more frequently by women receiving IPTp-AZCQ than those receiving IPTp-SP.	IPTp-AZCQ was not superior to IPTp-SP. The study was terminated earlier due to futility.

AQ: amodiaquine; AQSP: amodiaquine-sulfadoxine/pyrimethamine; IPTp: intermittent preventive treatment of malaria in pregnancy; LBW: low birth weight; MQ: mefloquine; RCT: randomized clinical trial; SP: sulfadoxine-pyrimethamine. CTX: cotrimoxazole; IPTp: intermittent preventive treatment of malaria in pregnancy; LBW: low birth weight; MQ: mefloquine; RCT: randomized clinical trial; SP: sulfadoxine-pyrimethamine. CTX: cotrimoxazole; DP: dihydroartemisinin-piperaquine; IPTp: intermittent preventive treatment of malaria in pregnancy; IST: intermittent screening and treatment; RCT: randomized clinical trial; SP: sulfadoxine-pyrimethamine. CQ: chloroquine; DP: dihydroartemisinin-piperaquine; IPTp: intermittent preventive treatment of malaria in pregnancy; LBW: low birthweight; RCT: randomized clinical trial; SP: sulfadoxine-pyrimethamine. AZ: azithromycin; AZCQ: azithromyicin/chloroquine; AZSP: azithromycin/sulfadoxine-pyrimethamine; IPTp: intermittent preventive treatment of malaria in pregnancy; LBW: low birthweight; RCT: randomized clinical trial; SP: sulfadoxine-pyrimethamine; w.: week.

## Data Availability

Not applicable.

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
