# Peer review of "Drugs for Intermittent Preventive Treatment of Malaria in Pregnancy: Current Knowledge and Way Forward"

_tropicalmed, 2022, doi:10.3390/tropicalmed7080152_

Round 1

Reviewer 1 Report

This article by Figuero-Romero is an authoritative review of different treatment options tested for intermittent preventive treatment of malaria in pregnancy. The review presents an excellent systematic review of this broad topic that is of substantial importance as a chemopreventive tool to prevent malaria in pregnant women. I found it very well written and very informative. This review will be of substantial interest to the malaria treatment and control community. I have only a few minor comments that require attention. 

1.     Section 2.7 mentions SP, mefloquine and chloroquine when discussing chemoprevention during the first trimester of pregnancy. It would be helpful to also mention pros and cons of dihydroartemisinin-pieraquine and also amodiaquine. 

2.     Line 290: Please write “proved” not “probed”

3.     Line 378: please spell out RTI

Author Response

We thank the reviewer for their comments. They all are very appropriate and will help us improve the final version of the article. We have provided a response for each of the comments/questions and we have modified the manuscript accordingly.

Reviewer 1

  1. Section 2.7. Mentions SP, mefloquine and chloroquine when discussion chemoprevention during the first trimester of pregnancy. It would be helpful to also mention pros and cons of dihydroartemisinin-piperaquine and also amodiaquine.
  2. Thank you for the suggestion. However, in this section we only discuss those drugs that could be used safely in first trimester of pregnancy.
  3. Line 290: Please write “proved” not “proved”.

Thank you for noting the error. We have corrected it.

  1. Line 378: please spell out RTI

The acronym (RTI: reproductive tract infections) has been spelt out before (line 254).

Reviewer 2 Report

This  paper is a review of drugs for intermittent preventive treatment of malaria by Dr Figueroa-Romero and colleagues from ISGlobal in Barcelona.  The paper is well written, and well laid out and makes a good contribution to the review literature.  I’ve added some comments below where I think that with minor additions of text the authors could make this into a much more important paper.

When we get onto a discussion of ‘what next’ there is one big issue that is missing from the paper.  The use of new drugs in pregnancy is something which is contemplated very conservatively.  So, if we imagine that a new chemical entity is approved for treatment in the next 5 years, it may be another 10 years or so before evidence is available for its activity in chemoprevention in pregnancy.  Monoclonal antibodies on the other hand, are generally thought to be safe (if they can demonstrate lack of binding to human placental and embryonic tissue), and do represent another way forwards.  This would be important to mention in the ‘future perspective’.  The vaccines (RTSS and R21) which the Barcelona and Manhica groups have worked on, have not been shown to be useful in pregnancy – and so the only hope in vaccination would be if the newer mRNA approaches show activity.  This may be worth (a somewhat negative) comment!

General comments

One of the great advantages of SP is the price.   A month’s coverage with SP costs $0.10, according to the global fund’s latest prices.  This compares to close to $2.00 for DP.  Do the authors have any perspective on whether procurement of newer more expensive drugs such as DP or mefloquine would be achievable?  This is somewhat reflected in the text – but the manuscript would be better if you actually gave the current GF prices, since these really do set the problem in context

Specific Comments

Section 1.2 The headline numbers are large; out of 33 million pregnancies, some 30% of mothers were infected with malaria, leading to 3% of children with low birth weight.  Are there any estimates from the group as to the direct impact of malaria in terms of mortality for either the mother or the fetus?  It might help also to translate the low birth weight into a more widely understood health impact metric such as DALY, to give the general audience more appreciation.

Section 1.3  The WHO recommendation of SP to be given twice during pregnancy is presumably a compromise situation.  Are there any data on whether better protection/efficacy is given by having monthly delivery during pregnancy – if not this is a gap that would be worth highlighting

There are data suggesting that the impact of SP in pregnancy is more than simply its effect on the malaria – hence the impact on LBW for SP appears to be better than DHA-PQP despite the superior antimalarial activity of the latter – this again is something that is worth highlighting.  It underlines just how difficult the search for ‘new treatments’ is, since the actual goals (target product profile) is not well defined

Section 2.1 Amodiaquine.  Although it has been traditional to make comments about the high frequency of adverse events with amodiaquine based on historical data, the reality in Africa appears to be different.  In the phase IV study in Ghana of amodiaquine-artesunate, the Sanofi-led team found only 3 cases of EPS in 15’000 patients; no hepatic signals or agraunocytosis have been reported from the use of SP-amodiaquine in children, despite over 45 million children being treated in 2021.

Section 2.3: Mefloquine.  It is worth adding that the US FDA has a black box warning for the neuropsychiatric effects of the drug.  The relatively high cost of mefloquine stems partly from this – that no manufacturer would want to assume the risk of making the drug.

Section 2.6 HIV infected pregnant women.  Could the authors add in what proportion of the 33 million pregnant women in SSA are HIV infected and on therapy, to help put things in perspective?   It would be good to spell out the increased risks of SP toxicity when given with co-trimoxazole as an antibacterial.  To the outsider the question could also be whether there are better drugs than co-trimoxazole for dealing with HIV related bacterial infections.  The observation that CTX gives a similar (noninferior) response to SP in terms of LBW is critical, since this also feeds into the hypothesis that the effects of SP are not purely as an anti-malarial.  This would be worth a little more discussion.

Section 2.7  It would be good to comment on what the total need in first trimester would be.  Presumably many women in Africa are not diagnosed as pregnancy in the first few weeks – so perhps give some indication from the authors experience as to whether the protection in the last few weeks of the first trimester (weeks 9-13) represents a significant unmet medical need. 

Atovaquone proguanil is curiously missing in this report – or it may just be that I have missed it.  There are many reasons why its not the ideal drug for IPTp, but these should at least be laid out.

Author Response

We thank the reviewer for their comments. They all are very appropriate and will help us improve the final version of the article. We have provided a response for each of the comments/questions and we have modified the manuscript accordingly.

Reviewer 2

This paper is a review of drugs for intermittent preventive treatment of malaria by Dr Figueroa-Romero and colleagues from ISGlobal in Barcelona. The paper is well written, and well laid out and makes a good contribution to the review literature. I’ve added some comments below where I think that with minor additions of text the authors could make this into a much more important paper.

R: Thank you for this comment and for acknowledging our contribution to the existing literature.

When we get onto a discussion of ‘what next’ there is one big issue that is missing from the paper. The use of new drugs in pregnancy is something which is contemplated very conservatively. So, if we imagine that a new chemical entity is approved for treatment in the next 5 years, it may be another 10 years or so before evidence is available for its activity in chemoprevention in pregnancy. Monoclonal antibodies on the other hand, are generally thought to be safe (if they can demonstrate lack of binding to human placental and embryonic tissue), and do represent another way forward. This would be important to mention in the ‘future perspective’. The vaccines (RTSS and R21) which the Barcelona and Manhica groups have worked on, have not been shown to be useful in pregnancy –and so the only hope in vaccination would be if the newer mRNA approaches show activity. This may be worth (a somewhat negative) comment!

R: We are grateful for this excellent suggestion. We have included a paragraph discussing other approaches for malaria control such as vaccines and monoclonal antibodies in the discussion and way forward section.

General comments

One of the great advantages of SP is the price. A month’s coverage with SP costs $0.10, according to the global fund’s latest prices. This compares to close to $2.00 for DP. Do the authors have any perspective on whether procurement of newer more expensive drugs such as DP or mefloquine would be achievable? This is somewhat reflected in the text – but the manuscript would be better if you actually gave the current GF prices, since these really do set the problem in context

R: Thank you for noting this. We have now included the price of a treatment course with SP vs with DP in the “discussion and way forward” section so it can put the reader in context.

Specific Comments

Section 1.2 The headline numbers are large; out of 33 million pregnancies, some 30% of mothers were infected with malaria, leading to 3% of children with low birth weight. Are there any estimates from the group as to the direct impact of malaria in terms of mortality for either the mother or the fetus? It might help also to translate the low birth weight into a more widely understood health impact metric such as DALY, to give the general audience more appreciation.

R: Thank you for suggesting this. We have included information about maternal mortality and stillbirths on section 1.2.

Clinical trials that assess the efficacy of antimalarials for IPTp use low birthweight as an endpoint to measure this efficacy, while DALYs are rarely used to reflect this, thus we preferred to use the same measures that we report in the following sections. DALYs caused by malaria in pregnancy depend on the life expectancy in each country, among other factors, which make this measure difficult to generalize for all SSA.

Section 1.3 The WHO recommendation of SP to be given twice during pregnancy is presumably a compromise situation. Are there any data on whether better protection/efficacy is given by having monthly delivery during pregnancy – if not this is a gap that would be worth highlighting

R: Thank you for your suggestion. Please note that current WHO guidelines recommend the uptake of monthly IPTp-SP doses, starting from the second trimester of pregnancy, at each scheduled ANC visit.  This recommendation was released in 2012 and it is based on evidence from different studies.  Among these studies, there is a meta-analysis (Kayentao et al. JAMA 2013), which included 7 controlled trials conducted in 5 sub-Saharan countries from 1994 to 2008, which showed that 3 or more doses (median of 4 doses) of IPTp with SP was superior to the standard 2 dose regimen in preventing LBW rates (relative risk reduction of 21% [95% CI 8-32]) both in HIV infected and uninfected pregnant women and in all gravidity groups. Furthermore, women who received a median of 4 doses of IPTp-SP compared to those on the 2-dose regimen also had a lower risk of moderate-severe maternal anemia, maternal malaria at delivery, and placental malaria. The meta-analysis, which included two trials in areas of Burkina Faso and Mali where the efficacy of SP remains high, showed that even in areas of high SP efficacy, 3 doses of SP were more effective than 2 doses.

There are data suggesting that the impact of SP in pregnancy is more than simply its effect on the malaria – hence the impact on LBW for SP appears to be better than DHA-PQ despite the superior antimalarial activity of the latter – this again is something that is worth highlighting. It underlines just how difficult the search for ‘new treatments’ is, since the actual goals (target product profile) is not well defined

R: We appreciate your comment and agree with you on the broad efficacy of SP. We have include a sentence in the discussion highlighting this issue (line 415).

Section 2.1 Amodiaquine. Although it has been traditional to make comments about the high frequency of adverse events with amodiaquine based on historical data, the reality in Africa appears to be different. In the phase IV study in Ghana of amodiaquine-artesunate, the Sanofi-led team found only 3 cases of EPS in 15’000 patients; no hepatic signals or agranulocytosis have been reported from the use of SP-amodiaquine in children, despite over 45 million children being treated in 2021.

R: Thank you for raising this point. We have include it in section 2.1 when presenting AQ safety and included the reference on the phase IV-Pharmacovigilance study conducted in Côte d’Ivoire which included over 12000 children (Assi et al. Malar J 2017). 

Section 2.3: Mefloquine. It is worth adding that the US FDA has a black box warning for the neuropsychiatric effects of the drug. The relatively high cost of mefloquine stems partly from this –that no manufacturer would want to assume the risk of making the drug.

R: We have included this side event in the Section 2.3.

Section 2.6 HIV infected pregnant women. Could the authors add in what proportion of the 33 million pregnant women in SSA are HIV infected and on therapy, to help put things in perspective?

R: Thank you for this suggestion. We have included available data from UNAIDS of the number of pregnant women with HIV and on antiretroviral therapy in 2020.

It would be good to spell out the increased risks of SP toxicity when given with co-trimoxazole as an antibacterial. To the outsider the question could also be whether there are better drugs than co-trimoxazole for dealing with HIV related bacterial infections. The observation that CTX gives a similar (noninferior) response to SP in terms of LBW is critical, since this also feeds into the hypothesis that the effects of SP are not purely as an anti-malarial. This would be worth a little more discussion.

R: Thank you for the suggestion. We have included some examples of sulfonamide-induced adverse drug reactions in the text. Additionally, we have expanded further the discussion on SP anti-bacterial activity. 

Section 2.7 It would be good to comment on what the total need in first trimester would be. Presumably many women in Africa are not diagnosed as pregnancy in the first few weeks – so perhaps give some indication from the authors experience as to whether the protection in the last few weeks of the first trimester (weeks 9-13) represents a significant unmet medical need.

R: Thank you for your comment. We have included the existing estimates on the proportion of women attending ANC services before 12th week of gestation in sub-Saharan Africa and the prevalence of malaria in first semester.

Atovaquone proguanil is curiously missing in this report – or it may just be that I have missed it. There are many reasons why it’s not the ideal drug for IPTp, but these should at least be laid out.

R: Thank you for pointing this out. We have only included in the review studies reporting on the efficacy and safety of drugs that have been evaluated for prevention of malaria in pregnancy as described in methods (Section 1.4). Atovaquone-proguanil has not been evaluated for prevention of malaria in pregnancy as it is not recommended for use in pregnancy due to limited data on safety.
